# Association Between Alexithymia, Sexual Dysfunctions, and Dyadic Adjustment in Obsessive–Compulsive Disorder

**DOI:** 10.3390/medicina61101802

**Published:** 2025-10-07

**Authors:** Tunahan Sun, Gonca Karakuş, Lut Tamam, Mehmet Emin Demirkol, Zeynep Namlı, Caner Yeşiloğlu

**Affiliations:** 1Department of Psychiatry, Düziçi State Hospital, 80600 Osmaniye, Turkey; 2Department of Psychiatry, Faculty of Medicine, Cukurova University, 01100 Adana, Turkey

**Keywords:** alexithymia, sexual dysfunction, dyadic adjustment, obsessive–compulsive disorder

## Abstract

*Background and Objectives*: Chronic mental disorders may negatively affect sexual functioning and dyadic adjustment. This study aimed to investigate the associations between alexithymia, sexual dysfunctions, and dyadic adjustment in patients with obsessive–compulsive disorder (OCD) and to compare these variables with those of healthy controls. *Materials and Methods*: This case–control study included 72 patients with OCD and 82 sociodemographically matched healthy controls. All participants completed the Toronto Alexithymia Scale (TAS-20), Arizona Sexual Experiences Scale (ASEX), Yale–Brown Obsessive Compulsive Scale (YBOCS), and Dyadic Adjustment Scale (DAS). Group comparisons were conducted using independent *t*-tests, Mann–Whitney U tests, and chi-square tests, while correlations were examined using Pearson’s analysis. *Results*: Patients with OCD had significantly higher TAS-20 scores (60.97 ± 11.15 vs. 43.18 ± 8.86, *p* < 0.001) and ASEX total scores (18.33 ± 4.93 vs. 13.76 ± 3.55, *p* < 0.001), alongside lower DAS scores (total and all subscales, *p* < 0.001) than controls. Within the OCD group, TAS-20 scores correlated positively with the total ASEX score (r = 0.366, *p* = 0.002) and negatively with the total DAS score (r = −0.339, *p* = 0.004) and subscales (all *p* < 0.05). *Conclusions*: Patients with OCD exhibit elevated alexithymia, greater sexual dysfunction, and reduced dyadic adjustment compared with healthy controls. Furthermore, alexithymia in patients with OCD is associated with impaired sexual functioning and dyadic adjustment. Assessing alexithymic traits and addressing them in treatment may improve social and familial functioning in this population.

## 1. Introduction

Obsessive–compulsive disorder (OCD) is a mental condition characterized by recurrent ritualistic behaviors or mental acts accompanied by undesired and compulsive thoughts, urges, and fantasies. OCD can severely impair daily routines, occupational functioning, social life, and interpersonal relationships, leading to a significant loss of functioning [1]. The lifetime prevalence of OCD is estimated to be between 1.1% and 3.5% [2]. It is often linked to other mental health conditions and disabilities [3]. Despite evidence-based treatments such as cognitive-behavioral therapy (CBT), pharmacotherapy, or their combination, relapse occurs in up to 59% of patients [4].

Alexithymia is defined as the inability to identify and articulate one’s emotions. It is characterized by a deficiency in the ability to identify the facial expressions of others, empathy, and emotion regulation [5]. In other words, alexithymia not only inhibits an individual’s ability to identify and express their own emotions and use them in daily life, but also prevents others from recognizing them [6]. Approximately 10% of the general population is affected, with a higher prevalence in men than in women [7]. Additionally, severe alexithymia was found in 45.7% of patients with psychiatric diagnoses, suggesting a possible role of alexithymia in the development and/or persistence of psychiatric disorders [8]. It has been suggested that OCD and alexithymia share a common neurobiological mechanism. It was reported that pathology in alexithymia was associated with regions in the ventromedial prefrontal cortex, including the orbitofrontal cortex and anterior cingulate cortex [9]. Some previous studies have reported that the ventromedial prefrontal cortex also plays an important role in patients with OCD [10]. Previous studies have reported high levels of alexithymia in patients with OCD [11,12] and that sexual/religious obsessions were positively associated with alexithymia [13]. However, research on this relationship is limited.

Sexual well-being is an indicator of quality of life [14]. Sexual dysfunctions (SDs) are defined as an impairment in the sexual response cycle that affects both men and women [15]. Previous research has revealed a connection between obsessive and compulsive symptoms and OCD diagnosis, lower sexual functionality, a higher incidence of SDs, and reduced sexual satisfaction [16]. Despite the potential for a substantial link between specific OCD symptoms and sexual functioning, only a limited number of studies have investigated the relationship between sexuality and OCD. Although SDs in patients with OCD have been attributed to the disorder itself or the medications in use, they are multifactorial and need to be investigated from different perspectives [17].

Alexithymia is another factor that may affect sexual function. Difficulties in recognizing and describing bodily sensations, such as arousal, negatively impact the quality of sexual interactions [18]. The limbic system controls sexuality, and the neocortex is associated with cognition and emotions. The neuropsychological model defines alexithymia as a disconnection between the neocortex and limbic system [19,20]. Studies on non-clinical samples have reported a negative correlation between alexithymia and sexual functioning [21,22].

Patients with OCD often face challenges in interpersonal communication, which can negatively impact their relationships with their partners [23]. Many conceal their symptoms to gain acceptance and avoid rejection, a behavior that reduces partner intimacy and contributes to poorer dyadic adjustment [24]. OCD has been consistently associated with family and marital distress [25]. A previous study found that patients with OCD reported lower satisfaction and intimacy with their partners [26].

Alexithymia is another important factor that influences dyadic adjustment. Individuals with high levels of alexithymia often lack the emotional skills required to establish and sustain intimate relationships [27]. Prior studies in non-clinical populations have reported negative correlations between alexithymia and dyadic adjustment [27,28].

However, the influence of alexithymia on SDs and dyadic adjustment, two factors critical to partner and family relationships, has not been thoroughly examined in OCD. Therefore, the present study primarily aimed to investigate alexithymia in patients with OCD and its association with SDs and dyadic adjustment. The secondary aim was to compare patients with OCD and healthy controls in terms of alexithymia, sexual functioning, and dyadic adjustment.

The following hypotheses guided this study:

**H1.** 
*Patients with OCD would exhibit higher levels of alexithymia, SDs, and lower dyadic adjustment than healthy controls.*


**H2.** 
*Among patients with OCD, the presence of alexithymia would be associated with greater SDs and lower dyadic adjustment.*


## 2. Materials and Methods

### 2.1. Participants

The present study was conducted at the outpatient clinics of the Department of Mental Health and Diseases, Balcali Hospital, Cukurova University Faculty of Medicine, Turkey. A total of 82 patients who were either married or had a regular sexual partner for at least three months were included, as assessed using the Structured Clinical Interview for the Diagnostic and Statistical Manual of Mental Disorders, Fifth Edition–Clinician Version (SCID-5-CV).

Inclusion criteria: Patients were eligible for inclusion if they were between 18 and 65 years of age, married or in a regular sexual relationship for at least three months, literate, and able to complete the study scales.

Exclusion criteria: Patients were excluded if they had comorbid psychiatric disorders, dementia, delirium, or intellectual disability associated with cognitive impairment, or if they were using psychotropic medications other than selective serotonin reuptake inhibitors (SSRIs). In addition, patients with medical conditions known to adversely affect sexual functioning (e.g., diabetes mellitus, cardiovascular diseases, endocrine disorders, genitourinary diseases) were excluded.

Based on information obtained from patients and their relatives, physical examination findings, and a review of clinical records, 10 patients were excluded due to a history of pelvic or abdominal surgery, chronic medical illnesses (such as diabetes mellitus, cardiovascular diseases, or those receiving hormone replacement therapy), or the use of medications other than SSRIs, as these conditions could adversely affect sexual functioning.

The control group included 82 volunteers with sociodemographic characteristics similar to those of the patients. Controls had no psychiatric disorders or complaints, no medical conditions, were not receiving any medication, and were either married or had a regular sexual partner for at least three months. The final sample consisted of 72 patients with OCD and 82 healthy controls.

### 2.2. Power Analysis

An a priori power analysis was conducted using G*Power 3.1 (Heinrich Heine University Düsseldorf, Düsseldorf, Germany). For a single group, the required minimum sample size was 51, and for two independent groups, the required minimum was 102 participants, assuming a medium effect size (Cohen’s d = 0.50), α = 0.05, and power (1−β) = 0.80. Our total sample of 154 (72 patients and 82 controls) exceeded this requirement. In addition, a post hoc power analysis was performed using G*Power 3.1.9.2 (Heinrich Heine University Düsseldorf, Düsseldorf, Germany). For the comparison of mean scores between the two groups, with sample sizes of 72 and 82, an effect size of d = 0.50, and α = 0.05, the achieved power was 92%. This further supports the robustness of our findings.

### 2.3. Procedure

In the first stage of the study, the first author conducted interviews lasting approximately 60 min with all participants. During these interviews, we collected sociodemographic data from the participants using a pre-prepared form. The diagnosis of OCD was confirmed using the DSM-5, and the SCID-5-CV was used to identify any comorbid psychiatric disorders accompanying OCD.

The Yale-Brown Obsessive Compulsive Scale (YBOCS) was used to assess OCD symptoms and the severity of the disorder. After the interview, the participants received information about the scales used. In the next stage, they were asked to complete the Toronto Alexithymia Scale (TAS-20) to evaluate their level of alexithymia, the Arizona Sexual Experiences Scale (ASEX) to assess sexual functioning, and the Dyadic Adjustment Scale (DAS) to evaluate dyadic adjustment. For patients who experienced difficulty completing the self-report scales, a clinician provided assistance and clarification on any points that were not understood.

### 2.4. Data Collection

#### 2.4.1. Sociodemographic and Clinical Data Form

The researchers developed this data form based on previous studies to gather specific information. It includes questions about marital status, age, gender, family history of mental disorders, educational background, employment status, smoking habits, alcohol and substance use, the number of hospitalizations, the duration of OCD, and the types of drug treatments received.

#### 2.4.2. Structured Clinical Interview for DSM-5-Clinician Version (SCID-5-CV)

It is a semistructured clinical interview guide aimed at assessing diagnoses defined by the DSM-5. It was developed by First et al. [29]. The SCID-5 was modified for the Turkish language, and a reliability study was conducted. The kappa coefficient in the Turkish version of the SCID-5 was very high and statistically significant, similar to previous SCID versions [30]. The scale was created as a clinical diagnostic instrument and completed by clinicians. The SCID-5-CV was used in our study to investigate comorbid psychiatric diagnoses and to confirm the diagnosis of patients with OCD.

#### 2.4.3. Yale–Brown Obsessive Compulsive Scale (YBOCS)

It is a clinician-administered scale aimed at assessing the severity and type of obsession and compulsion. Karamustafalıoğlu et al. assessed the reliability and validity of the Turkish version of the scale [31]. Although it contains 19 items, only the first 10 items are used to determine the total score. Accordingly, items 1–5 indicate the degree of obsession, and items 6–10 indicate the degree of compulsion.

#### 2.4.4. Toronto Alexithymia Scale (TAS-20)

The TAS-20 is a 20-item, 5-point Likert-type self-report scale. Higher scores on the scale indicate difficulty in expressing emotions. Güleç et al. studied the reliability and validity of the Turkish version [32]. In the present study, participants who scored ≥61 points were classified as alexithymic, while those who scored <61 were classified as non-alexithymic.

#### 2.4.5. Arizona Sexual Experiences Scale (ASEX)

This scale was developed by McGahuey et al. to quickly and easily detect sexual complications [33]. There are two distinct forms, one for men and one for women. It is a 6-point Likert-type self-report scale consisting of five items. Satisfaction with orgasm, sexual desire, ability to reach orgasm, vaginal lubrication/penile erection, and arousal were assessed. Higher scores indicate higher SDs. The overall score ranged from 5 to 30. A score of ≥5 for any item or a total scale score of ≥17 indicates that the person has SDs [16,34]. Soykan conducted a study on the Turkish reliability and validity of the ASEX [35]. In the present study, participants with a score of ≥5 on any item or a total scale score of ≥17 were considered individuals with SDs, and others without SDs.

#### 2.4.6. Dyadic Adjustment Scale (DAS)

Spanier developed this scale to evaluate the quality of relationships among married or cohabiting partners [36]. The DAS is a self-reported scale that contains 32 items and is divided into four subdomains: Affectional Expression, Dyadic Cohesion, Dyadic Satisfaction, and Dyadic Consensus. Higher DAS scores indicate better relationship quality and greater dyadic adjustment. Fisıloğlu and Demir conducted a study assessed the reliability and validity of the Turkish version of the DAS [37].

### 2.5. Ethics

Ethics committee approval for the study was obtained on 4 September 2020, from the Cukurova University Faculty of Medicine, Non-Interventional Clinical Research Ethics Committee, upon meeting no. 103. After being fully briefed on the study, all participants provided their written informed consent. This study was conducted in accordance with the guidelines of the Declaration of Helsinki.

### 2.6. Statistical Analysis

All statistical analyses were performed using IBM SPSS Statistics version 25 (IBM Corp., Armonk, NY, USA). Statistical significance was set at *p* < 0.05. Normality was assessed by examining skewness and kurtosis values (acceptable range: −1.5 to +1.5) and by visually inspecting the histogram plots. For normally distributed variables, independent samples *t*-tests were applied, and the results are presented as mean ± standard deviation (SD). For non-normally distributed variables, the Mann–Whitney U test was used, and the results are expressed as median and interquartile range. Pearson’s correlation analysis was conducted to examine the relationships between normally distributed numerical variables. For categorical variables, chi-square tests were used; Fisher’s exact test was applied when the expected frequency was <5, and Yates’ continuity correction was used when the expected frequencies ranged between 5 and 25.

A flow chart summarizing the study design, eligibility assessment, participant selection, applied measures, subgroup comparisons, and statistical analyses is presented in Figure 1.

## 3. Results

### Figures, Tables and Schemes

There were no significant differences between patients with OCD and healthy controls in terms of age, sex distribution, education duration, length and type of marriage, marital status, smoking, alcohol or substance use, and place of residence (all *p* > 0.05). However, employment status differed significantly, with 56.9% of patients with OCD (*n* = 41) employed compared to 87.8% of controls (*n* = 72; *p* < 0.001). In addition, a family history of mental disorders was reported by 45.8% of OCD patients (*n* = 33) versus 22% of controls (*n* = 18; *p* = 0.03). The detailed distributions of the sociodemographic variables in both groups are presented in Table 1.

The mean YBOCS scores of patients with OCD were 10.61 ± 3.22 for the obsession subscale, 10.85 ± 2.99 for the compulsion subscale, and 21.46 ± 6.02 for the total score. The TAS-20 total scores were significantly higher in the OCD group than in the control group (60.97 ± 11.15 vs. 43.18 ± 8.86, *p* < 0.001). Patients with OCD also had significantly higher ASEX scores across all subscales: lubrication/erection (*p* < 0.001), arousal (*p* < 0.001), desire (*p* < 0.001), orgasm (*p* = 0.01), and satisfaction (*p* < 0.001), as well as the total ASEX score (*p* < 0.001). In contrast, the DAS total and subscale median scores were significantly higher in the control group than in the OCD group (all *p* < 0.001). Detailed comparisons of the scale scores between the groups are presented in Table 2.

The median and quartile values of the YBOCS obsession and compulsion subscales, as well as the YBOCS total score and ASEX total score, were significantly higher in alexithymic OCD patients than in their non-alexithymic counterparts (all *p* < 0.001). Conversely, DAS subscale and total scores were significantly higher among non-alexithymic OCD patients than among alexithymic patients (all *p* < 0.05). The detailed group comparisons are presented in Table 3.

The total YBOCS and YBOCS compulsion subscale scores were significantly higher in patients with OCD with SDs than in those without (*p* = 0.01 and *p* = 0.003, respectively). Although not statistically significant, patients with SDs had higher obsession subscale scores (*p* = 0.051). No significant differences were observed between the two groups in terms of sociodemographic variables, including age, sex, education, marital status, duration and type of marriage, family living situation, occupation, place of residence, or duration of OCD (all *p* > 0.05). In contrast, DAS subscale and total scores were significantly lower in patients with SDs than in those without SDs (all *p* < 0.05). Moreover, the TAS-20 scores were significantly higher in patients with SDs (*p* = 0.02). A detailed comparison between patients with and without SDs is presented in Table 4.

Pearson’s correlation analysis showed that TAS-20 scores were positively associated with ASEX (r = 0.366, *p* = 0.002) and negatively with DAS (r = −0.339, *p* = 0.004). YBOCS obsession scores correlated positively with TAS-20 (r = 0.304, *p* = 0.009), while YBOCS compulsion scores correlated positively with TAS-20 (r = 0.336, *p* = 0.004) and ASEX (r = 0.266, *p* = 0.024), but negatively with DAS (r = −0.307, *p* = 0.009). YBOCS total scores were also positively associated with TAS-20 (r = 0.330, *p* = 0.005) and ASEX (r = 0.256, *p* = 0.030), and negatively with DAS (r = −0.275, *p* = 0.019). Finally, ASEX scores showed a moderate negative correlation with DAS (r = −0.442, *p* < 0.001). No other significant correlations were found. Table 5 summarizes these intercorrelations.

## 4. Discussion

To the best of our knowledge, this is the first study to compare patients with OCD and healthy controls in terms of alexithymia, SDs, and dyadic adjustment. The most significant finding of our study was that patients with OCD and alexithymia exhibited greater impairment in sexual functioning and lower dyadic adjustment than those without alexithymia. Additionally, we found that patients with OCD had higher alexithymia scores, poorer sexual functioning, and lower levels of dyadic adjustment than the healthy population.

In the present study, the levels of alexithymia in patients with OCD were significantly higher than those in the healthy controls. This finding is consistent with previous studies that also found that patients with OCD exhibited greater levels of alexithymia than healthy individuals [11,12,38]. These findings indicate that patients with OCD struggle to recognize and identify their emotions, suggesting that chronic OCD may contribute to alexithymic traits.

In the present study, alexithymic patients with OCD had significantly more severe obsessive and compulsive symptoms than non-alexithymic patients. Additionally, positive correlations were observed between obsession and compulsion severity and alexithymia levels. These findings can be interpreted in two ways: either alexithymia increases as the severity of OCD increases, or individuals with alexithymic traits experience more severe clinical manifestations of OCD. Khosravani et al. reported that OCD symptoms were more severe in patients with alexithymia than in those without alexithymia [38]. De Berardis et al. found that symptoms were more severe in patients with OCD who had higher levels of alexithymia [39]. Roh et al. found a positive correlation between alexithymia, obsession levels, and severity of OCD symptoms [13]. Carpenter and Chung found that alexithymia in patients with OCD correlates with the severity and number of OCD symptoms [40]. Uslu et al. found no significant correlation between the severity of obsessive symptoms and alexithymia, which was attributed to the small sample size of their study [12]. The results of the literature review and present study align with each other. These findings indicate that CBT aimed at addressing alexithymia may have a positive impact on the progression of the disease by decreasing the severity of OCD symptoms.

In the present study, patients with OCD exhibited significantly greater impairments in arousal, sexual desire, orgasm, penile erection, and satisfaction with orgasm than healthy controls. The higher rate of SDs in patients with OCD than in the control group supports the findings of previous studies in the relevant literature, which compared patients with OCD with the control group [16,34]. Many patients with OCD are dissatisfied with their sexual relationships and tend to avoid sexual intercourse [41]. Some cognitive biases associated with OCD may also affect sexual satisfaction and sexual functioning. Additionally, certain symptoms such as contamination can impair sexual performance [42]. Sexual obsessions can cause feelings of guilt in some patients. Additionally, drug treatment may lead to SDs [43]. The results of this study may be influenced by the nature of OCD or the medication used. However, the impact of alexithymia and dyadic adjustment on sexuality should not be overlooked.

In the present study, patients with OCD and comorbid SDs demonstrated significantly higher severity of compulsions and OCD symptoms than those without SDs. Although patients with OCD and SDs showed greater obsession severity, this difference was not statistically significant. Furthermore, the study found that the incidence of SDs increased with the severity of OCD symptoms and compulsions in these patients. This aligns with the findings of Thakurta et al., who reported a positive correlation between the severity of OCD symptoms and the presence of SDs [44]. Raisi et al. found a negative correlation between OCD symptom severity and sexual satisfaction [45]. OCD is a chronic disorder caused by impairments in several areas of function [46], and the results of the present study indicate that as the severity of the disease increases, the degree of impairment in dyadic relationships and sexuality, which are important areas of function, also increases. Compulsions, such as repeated cleaning after sex and extensive cleaning efforts, can take a long time and are often noticeable. These behaviors can lead to sexual avoidance and development of SDs.

In the present study, patients with OCD and comorbid SDs showed significantly greater alexithymia than their counterparts without SDs. Additionally, alexithymic patients with OCD demonstrated greater SDs than their non-alexithymic counterparts. A positive correlation was observed between the level of alexithymia and the severity of SDs in patients with OCD. This finding suggests that as the level of alexithymia increases in patients with OCD, sexual function declines. However, to the best of our knowledge, no study has specifically examined the relationship between alexithymia and SDs in patients with OCD. Giuliani et al. conducted a study on individuals without mental disorders and revealed that alexithymia levels were significantly higher in those with SDs than in a control group without SDs [21]. Berenguer et al. found that higher levels of alexithymia in women were linked to poorer functioning across all dimensions of sexual function, except for the sexual desire stage [22]. Wise et al. reported no relationship between types of SDs and levels of alexithymia [47]. The association between alexithymic individuals and sexual difficulties seems intuitively logical. In human sexuality, the ability to express and experience emotions is one of the most important elements [48]. While further research is needed to determine whether alexithymia is associated with SDs, especially in clinical populations such as those with OCD, clinicians must be more aware of alexithymic traits in patients with OCD.

The present study also examined how the participants interpreted their dyadic adjustment, in addition to sexual functions reflecting partner relationships. The dyadic adjustment levels of patients with OCD were significantly lower than those of healthy controls. Additionally, there was a negative relationship between obsession levels, compulsion levels, and the severity of OCD symptoms and dyadic adjustment in patients with OCD. Given these results, it can be suggested that an increase in the severity of OCD symptoms decreases dyadic adjustment. A previous study reported a negative correlation between the severity of obsessions and relationship satisfaction [49]. Another study suggested that couples experienced less marital distress as OCD symptoms decreased after exposure treatment, and that there was a negative relationship between OCD severity and relationship satisfaction [50]. Riggs et al. reported a significant increase in marital adjustment after CBT in patients with OCD who were dissatisfied with their marriages before treatment and reported that the treatment of OCD symptoms contributed to improvements in marital satisfaction [25]. OCD significantly impacts issues related to marriage and sexuality, as it causes excessive anxiety and strains relationships. As obsessions become more severe, patients with OCD, who are more focused on intrusive thoughts, spend less time and energy on intimate relationships with their partners [25,50]. This can decrease dyadic adjustment, as expected. Family- and couple-based treatment approaches should also be considered in the follow-up and treatment of patients diagnosed with OCD to break the vicious cycle, where severe clinical symptoms negatively affect dyadic adjustment, and low dyadic adjustment, in turn, worsens the severity of the disorder.

In the present study, a negative relationship was found between dyadic adjustment and alexithymia levels in the patient group; dyadic adjustment was significantly lower among alexithymic OCD patients than among non-alexithymic patients. These results confirm that the occurrence of alexithymic traits in patients with OCD decreases dyadic adjustment. The authors of the present study found no previous studies that assessed alexithymia and dyadic adjustment in patients with OCD. Previous studies with individuals without mental disorders reported lower dyadic adjustment in individuals with high levels of alexithymia. Aghighi et al. reported a negative relationship between alexithymia and dyadic adjustment and a significant positive relationship between alexithymia and marital stress [51]. Yelsma and Marrow reported that difficulties in couples’ emotional expression in marriage reduce both their own and their spouses’ marital satisfaction [52]. Other studies also reported significant negative correlations between alexithymia and dyadic adjustment [27,28,53]. OCD patients with higher levels of alexithymia may struggle to establish and maintain intimate relationships.

Taylor et al. suggested that alexithymia was not a predictor of dyadic adjustment [54]. Accordingly, the present study also examined sexual functioning, along with disease severity and alexithymia, to identify variables related to dyadic adjustment in patients with OCD. One of the more important factors influencing dyadic adjustment is sexual functioning. Impaired sexual functioning negatively impacts dyadic adjustment; conversely, dyadic adjustment can affect sexual functioning [55].

In the present study, dyadic adjustment was higher in OCD patients without SDs compared to those with SDs, and a negative association was observed between SDs and dyadic adjustment. Raeisi et al. reported a significant negative relationship between SDs and marital satisfaction in patients with OCD [56]. Similar to our study, in a study involving patients diagnosed with bipolar disorder, dyadic adjustment was lower in those who had SDs than in those without SDs [57]. These data support the bidirectional relationship between SDs and impaired dyadic adjustment in patients with OCD and similar chronic mental disorders.

In the present study, the employment rate of patients with OCD was significantly lower than that of the control group. OCD is a chronic mental disorder that negatively impacts a person’s family, social, and work functioning [46]. This might be caused by the loss of function related to OCD.

### 4.1. Strengths

To the best of our knowledge, this is the first study to examine the interaction of alexithymia, SDs, and dyadic adjustment simultaneously in patients with OCD, whereas previous research has typically explored these variables only in pairs. The inclusion of a sociodemographically matched healthy control group strengthens the validity of the findings by allowing direct comparisons between groups. Furthermore, an a priori power analysis was conducted, and the sample size was sufficient to detect statistically significant differences.

### 4.2. Limitations

The present study has several limitations. The cross-sectional design prevents us from establishing cause-and-effect relationships between the variables. Although we met the required sample size based on the power analysis, the relatively small sample size remains a significant limitation. Future studies with larger sample sizes may produce more statistically significant results. Furthermore, the medications used by the patient group make it difficult to differentiate between medication-induced SDs and SDs related to OCD. Additionally, the effects of pharmacological treatments on sexual function were evaluated through self-reports, which might not yield the most accurate results. More objective data could be gathered by including assessments such as hormone measurements and blood prolactin levels. Another important limitation of this study is its questionnaire-based design. While standardized self-report instruments such as TAS-20, ASEX, and DAS provide reliable and validated measures, they may not fully capture the complexity and multidimensionality of human behavior, emotional awareness, and dyadic relationships. Responses are influenced by how participants interpret the questions, their level of self-awareness, and their willingness to disclose sensitive information. Moreover, questionnaires tend to simplify complex real-world problems into predefined categories, which can limit the depth of understanding. In the case of alexithymia in particular, it is difficult to determine whether an individual is truly alexithymic or whether their responses reflect subjective perceptions of emotional difficulties. Therefore, the findings should be interpreted with caution, acknowledging that questionnaire-based methods can only provide a partial representation of the underlying psychological and interpersonal processes. In addition, dyadic adjustment was assessed solely from the patients’ perspective, and partners’ views were not included. Future studies incorporating partner reports could provide a more comprehensive understanding of dyadic adjustment. Finally, the control group was restricted to individuals without medical or psychiatric conditions or medication use, which may limit the generalizability of the findings to the broader population.

## 5. Conclusions

This study demonstrates that patients with OCD exhibit significantly higher levels of alexithymia and SDs, together with lower dyadic adjustment, compared to healthy controls. Furthermore, the presence of alexithymia in OCD patients was associated with poorer sexual functioning and reduced dyadic adjustment, highlighting its potential role as a critical factor in interpersonal and relational difficulties. These findings emphasize the importance of evaluating alexithymic traits in OCD and addressing their impact on sexual and relational functioning in clinical practice. Incorporating such assessments into treatment planning may facilitate more comprehensive interventions and improve patients’ social and familial functioning.

Future research should include longitudinal studies with larger and more diverse samples, particularly among newly diagnosed patients, and consider stratification by medication status. Additionally, more research is needed to explore the connections between alexithymia, SDs, and dyadic adjustment in other psychiatric disorders to understand their overall clinical importance better.

## Figures and Tables

**Figure 1 medicina-61-01802-f001:**
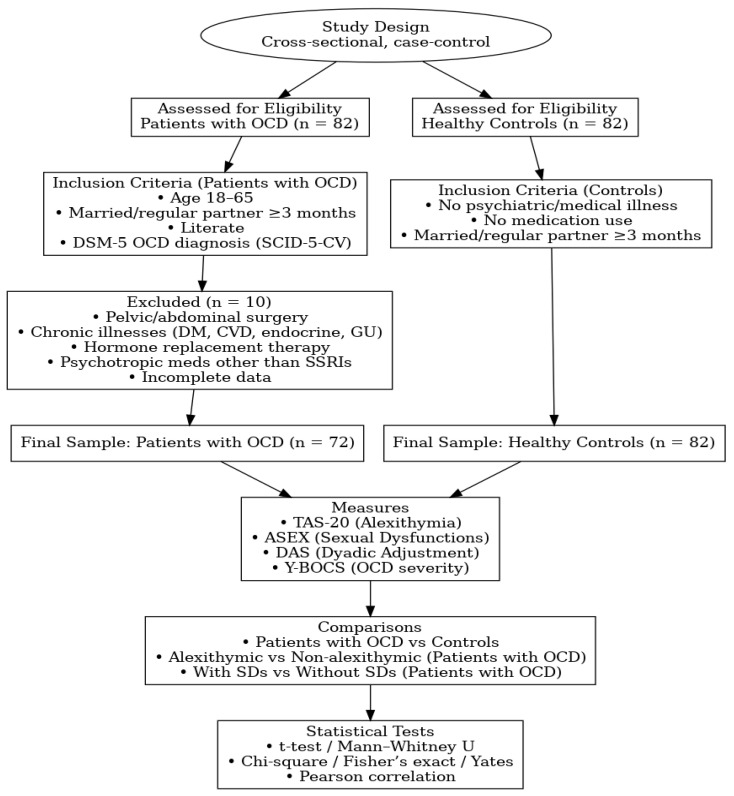
Flow chart of study design, participant selection, measures, comparisons, and statistical analyses. Abbreviations: TAS-20: Toronto Alexithymia Scale, ASEX: Arizona Sexual Experience Scale, DAS: Dyadic Adjustment Scale, Y-BOCS: Yale-Brown Obsessive Compulsive Scale, SDs: Sexual Dysfunctions, SSRIs: Selective Serotonin Reuptake Inhibitors, DM: Diabetes Mellitus; CVD: Cardiovascular Diseases, GU: Genitourinary.

**Table 1 medicina-61-01802-t001:** Sociodemographic variables of participants.

	Control	OCD		
	Mean ± SD	Mean ± SD	t	*p*-Value
Age	36.96 ± 8.08	36.67 ± 8.82	0.22	0.83 *
Average years of education	13.39 ± 4.04	13.01 ± 4.79	0.53	0.60 *
Duration of marriage (years)	10.83 ± 10.10	12.38 ± 9.93	−0.96	0.34 *
	*n* (%)	*n* (%)	χ^2^	*p*-value
Sex			0.05	0.83 ^a^
Female	43 (52.4)	39 (54.2)		
Male	39 (47.6)	33 (45.8)		
Marital status			0.92	0.34 ^b^
Single	16 (19.5)	9 (12.5)		
Married	66 (80.5)	63 (87.5)		
Type of marriage			0.87	0.35 ^b^
Companionate marriage	45 (68.2)	37 (58.7)		
Arranged marriage	21 (31.8)	26 (41.3)		
Type of family			2.13	0.34 ^b^
Nucleus family	65 (79.3)	61 (84.7)		
Extended family	7 (8.5)	7 (9.7)		
Living alone	10 (12.2)	4 (5.6)		
Profession			17.14	**<0.001 ^b^**
Not employed	10 (12.2)	31 (43.1)		
Employed	72 (87.8)	41 (56.9)		
Place of residence			1.07	0.30 ^b^
Smaller than the provincial center	8 (9.8)	12 (16.7)		
Provincial center	74 (90.2)	60 (83.3)		
Family history of mental disorders			8.82	**0.003 ^b^**
No	64 (78)	39 (54.2)		
Yes	18 (22)	33 (45.8)		
Smoking			0.46	0.50 ^a^
No	50 (61)	40 (55.6)		
Yes	32 (39)	32 (44.4)		
Alcohol			0.06	*p* = 0.81 ^b^
No	59 (72)	54 (75)		
Yes	23 (28)	18 (25)		
Substance				1 ^c^
No	80 (97.6)	71 (98.6)		
Yes	2 (2.4)	1 (1.4)		

*: Independent *t*-test, χ^2^ = Pearson’s chi-square test, ^a^: chi-square analysis, ^b^: Yates’ continuity correction, ^c^: Fisher’s exact test, Mean ± SD: Mean value and standard deviations. Statistically significant differences are highlighted in bold.

**Table 2 medicina-61-01802-t002:** A comparison of participants’ scale scores.

	Control	OCD			
	Mean ± SD	Mean ± SD	t		*p*-Value
TAS-20	43.18 ± 8.86	60.97 ± 11.15	−11.02		**<0.001 ***
Sexual desire	2.73 ± 1.10	3.92 ± 1.31	−6.10		**<0.001 ***
Arousal	2.77 ± 1.06	3.88 ± 1.40	−5.57		**<0.001 ***
Lubrication/Erection	2.67 ± 0.93	3.64 ± 1.27	−5.33		**<0.001 ***
Orgasm	3.30 ± 0.88	3.74 ± 1.21	−2.50		**0.01 ***
Satisfaction from orgasm	2.28 ± 0.91	3.17 ± 1.15	−5.34		**<0.001 ***
ASEX Total Score	13.76 ± 3.55	18.33 ± 4.93	−6.67		**<0.001 ***
	M (1Q–3Q)	M (1Q–3Q)	U	Z	
Dyadic Consensus	52(47.75–59)	41(33.25–49.75)	1333	−5.87	***p* < 0.001 ****
Dyadic Satisfaction	39(33.5–44)	30(26–37)	1431.5	−5.51	***p* < 0.001 ****
Affectional Expression	10(8–11)	7(4–9)	1169.5	−6.50	***p* < 0.001 ****
	Mean ± SD	Mean ± SD	t		
Dyadic Cohesion	16.80 ± 4.24	12.22 ± 4.95	6.19		**<0.001 ***
	M (1Q–3Q)	M (1Q–3Q)	U	Z	
DAS Total Score	119.5(106.75–127)	91.5(74.25–108.75)	1108	−6.68	**<0.001 ****

*: Independent *t*-test, **: Mann–Whitney U analysis, TAS-20: Toronto Alexithymia Scale Total Score, ASEX: Arizona Sexual Experiences Scale, DAS: Dyadic Adjustment Scale, M (1Q–3Q): Median and Quartile Values, Mean ± SD: Mean and Standard Deviation. Statistically significant differences are highlighted in bold.

**Table 3 medicina-61-01802-t003:** Scale scores of patients with OCD with and without alexithymia.

	Alexithymia+	Alexithymia−	U	Z	*p*-Value
	M (1Q–3Q)	M (1Q–3Q)			
YBOCS obsession	11(10–13)	0(0–5.75)	711	−7.52	**<0.001 ****
YBOCS compulsion	11.5(10–14)	0(0–5)	632.5	−7.86	**<0.001 ****
YBOCS Total Score	22.5(20–26.25)	0(0–10)	663.5	−7.70	**<0.001 ****
ASEX Total Score	19.5(16.75–21.75)	14(12–16.75)	1028	−5.76	**<0.001 ****
	Mean ± SD	Mean ± SD	t		*p*-value
Dyadic Consensus	38.23 ± 10.88	45.62 ± 11.59	−2.75		**0.01 ***
Dyadic Satisfaction	27.53 ± 7.56	34.14 ± 9.25	−3.32		**0.001 ***
Affectional Expression	5.72 ± 2.51	7.76 ± 2.69	−3.28		**0.002 ***
Dyadic Cohesion	11 ± 4.47	14.03 ± 5.14	−2.66		**0.01 ***
DAS Total Score	81.98 ± 21.73	101.44 ± 25.79	−3.46		**0.001 ***

*: Independent *t*-test, **: Mann–Whitney U analysis, YBOCS: Yale–Brown Obsessive Compulsive Scale, DAS: Dyadic Adjustment Scale, ASEX: Arizona Sexual Experiences Scale, Mean ± SD: Mean and Standard Deviation, M (1Q–3Q): Median and Quartile Values. Statistically significant differences are highlighted in bold.

**Table 4 medicina-61-01802-t004:** Comparison of sociodemographic variables and clinical scale scores in OCD patients with and without SDs.

	SDs+	SDs−	t	*p*-Value
	Mean ± SD	Mean ± SD		
Age	35.83 ± 7.99	38.24 ± 10.20	−1.11	0.27 *
Years in education	13.23 ± 5.23	12.60 ± 3.87	0.53	0.60 *
Duration of marriage	11.77 ± 9.26	13.52 ± 11.19	−0.71	0.48 *
Duration of OCD (months)	145.68 ± 110.44	111.16 ± 100.75	1.30	0.20 *
YBOCS obsession	11.15 ± 3.25	9.60 ± 2.97	1.98	0.051 *
YBOCS compulsion	11.60 ± 2.78	9.44 ± 2.93	3.08	**0.003 ***
YBOCS Total Score	22.74 ± 5.74	19.04 ± 5.88	2.59	**0.01 ***
Dyadic Consensus	37.87 ± 11.25	47.48 ± 9.92	−3.59	**0.001 ***
Dyadic Satisfaction	26.85 ± 8.13	36.48 ± 6.46	−5.12	**<0.001 ***
Affectional Expression	5.66 ± 2.61	8.20 ± 2.24	−4.12	**<0.001 ***
Dyadic Cohesion	10.89 ± 4.63	14.72 ± 4.62	−3.34	**0.001 ***
DAS Total Score	80.81 ± 23.11	106.76 ± 19.87	−4.75	**<0.001 ***
TAS-20	63.17 ± 10.26	56.84 ± 11.76	2.37	**0.02 ***
	*n* (%)	*n* (%)	χ^2^	
Sex			1.03	0.31 ^b^
Female	28 (59.6)	11 (44)		
Male	19 (40.4)	14 (56)		
Marital status				0.71 ^c^
Single	5 (10.6)	4 (16)		
Married	42 (89.4)	21 (84)		
Type of marriage			0.21	0.65 ^b^
Companionate marriage	26 (61.9)	11 (52.4)		
Arranged marriage	16 (38.1)	10 (47.6)		
Type of family				1 ^c^
Nucleus family	39 (83)	22 (88)		
Extended family	5 (10.6)	2 (8)		
Lives alone	3 (6.4)	1 (4)		
Profession			0.02	0.90 ^b^
Not employed	21 (44.7)	10 (40)		
Employed	26 (55.3)	15 (60)		
Place of residence				0.20 ^c^
Smaller than the provincial center	10 (21.3)	2 (8)		
Provincial center	37 (78.7)	23 (92)		

χ^2^ = Pearson’s chi-square test, ^b^: Yates’ continuity correction, ^c^: Fisher’s exact test, *: independent *t*-test, YBOCS: Yale–Brown Obsessive-Compulsive Scale, DAS: Dyadic Adjustment Scale, TAS-20: Toronto Alexithymia Scale Total Score, Mean ± SD: Mean and Standard Deviation, SDs: Sexual Dysfunctions. Statistically significant differences are highlighted in bold.

**Table 5 medicina-61-01802-t005:** Correlations of scale scores in patients with OCD.

	YBOCS-O	YBOCS-C	YBOCS-T	ASEX-T	TAS-20	DAS-T
**YBOCS-O**						
r	1	0.872	0.970	0.231	**0.304 ****	−0.228
*p*		<0.001	<0.001	0.051	**0.009**	0.054
**YBOCS-C**						
r		1	0.965	**0.266 ***	**0.336 ****	**−0.307 ****
*p*			<0.001	**0.024**	**0.004**	**0.009**
**YBOCS-T**						
r			1	**0.256***	**0.330 ****	**−0.275 ***
*p*				**0.030**	**0.005**	**0.019**
**ASEX-T**						
r				1	**0.366 ****	**−0.442 *****
*p*					**0.002**	**<0.001**
**TAS-20**						
r					1	**−0.339 ****
*p*						**0.004**
**DAS-T**						
r						1
*p*						

Pearson correlation analysis *: *p* < 0.05, **: *p* < 0.01, ***: *p* < 0.001, YBOCS-O: Yale–Brown Obsessive Compulsive Scale–Obsession Subscale, YBOCS-C: Yale–Brown Obsessive Compulsive Scale–Compulsion Subscale, YBOCS-T: Yale–Brown Obsessive Compulsive Scale–Total Score, ASEX-T: Arizona Sexual Experiences Scale Total Score, TAS-20: Toronto Alexithymia Scale Total Score, DAS-T: Dyadic Adjustment Scale Total Score. Statistically significant differences are highlighted in bold.

## Data Availability

The data are available on request from the corresponding author. The data are not publicly available due to ethical restrictions and privacy concerns involving human participants.

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
