# Peer review of "Association Between Alexithymia, Sexual Dysfunctions, and Dyadic Adjustment in Obsessive–Compulsive Disorder"

_medicina, 2025, doi:10.3390/medicina61101802_

Round 1
Reviewer 1 Report (Previous Reviewer 2)
Comments and Suggestions for Authors
This study showed that obsessive-compulsive disorder is likely to be accompanied by alexithymia and sexual dysfunction. This comparison with healthy controls provides useful information when considering support for OCD patients.
I don't think there's much change from the previous version.
If I were to give advice on areas for improvement, it would be to write the discussion section better. In research like this, taking into account the differences between the questionnaire and reality will increase objectivity.
Questionnaire-based research involves a designed framework: responses are obtained according to pre-defined questions, and the scope and content of the survey are determined by the questionnaire. As in this case, responses are collected and analyzed as numerical data, making them suitable for objectively identifying specific trends and characteristics. In contrast, real-world problems are complex and multifaceted: human behavior, psychology, and situations are highly complex and diverse, and a questionnaire can only capture a small portion of these elements. Therefore, since it is impossible to determine whether an OCD patient is actually suffering from alexithymia, it is best to carefully describe this aspect. In other words, responses are greatly influenced by how respondents understand the questions and the wording they use. Furthermore, respondents may only be able to answer aspects that they are aware of. It may be even better to write an analysis that takes into account the limitations of discussing alexithymia solely through a questionnaire, as it can be difficult to grasp context and deeper meanings.
Author Response
Please see the attachment.
Reviewer-1
This study showed that obsessive-compulsive disorder is likely to be accompanied by alexithymia and sexual dysfunction. This comparison with healthy controls provides useful information when considering support for OCD patients. I don't think there's much change from the previous version.
- If I were to give advice on areas for improvement, it would be to write the discussion section better. In research like this, taking into account the differences between the questionnaire and reality will increase objectivity.
Questionnaire-based research involves a designed framework: responses are obtained according to pre-defined questions, and the scope and content of the survey are determined by the questionnaire. As in this case, responses are collected and analyzed as numerical data, making them suitable for objectively identifying specific trends and characteristics. In contrast, real-world problems are complex and multifaceted: human behavior, psychology, and situations are highly complex and diverse, and a questionnaire can only capture a small portion of these elements. Therefore, since it is impossible to determine whether an OCD patient is actually suffering from alexithymia, it is best to carefully describe this aspect. In other words, responses are greatly influenced by how respondents understand the questions and the wording they use. Furthermore, respondents may only be able to answer aspects that they are aware of. It may be even better to write an analysis that takes into account the limitations of discussing alexithymia solely through a questionnaire, as it can be difficult to grasp context and deeper meanings
- We sincerely thank the reviewer for this thoughtful and constructive suggestion. We agree that questionnaire-based research has inherent limitations, as responses are influenced by participants' interpretations of the questions, their level of awareness, and the limitations of self-report tools in fully capturing the complexity of real-life behaviors and psychological experiences. In line with this comment, we have expanded the Discussion section to acknowledge these limitations more explicitly. Specifically, we emphasized that while standardized questionnaires (TAS-20, ASEX, DAS) allow for objective measurement and comparability, they may not fully reflect the multidimensional nature of alexithymia, sexual dysfunction, and dyadic adjustment in OCD patients.
“4.2. Limitations
The present study has several limitations. The cross-sectional design prevents us from establishing cause-and-effect relationships between the variables. Although we met the required sample size based on the power analysis, the relatively small sample size remains a significant limitation. Future studies with larger sample sizes may produce more statistically significant results. Furthermore, the medications used by the patient group make it difficult to differentiate between medication-induced SDs and SDs related to OCD. Additionally, the effects of pharmacological treatments on sexual function were evaluated through self-reports, which might not yield the most accurate results. More objective data could be gathered by including assessments such as hormone measurements and blood prolactin levels. Another important limitation of this study is its questionnaire-based design. While standardized self-report instruments such as TAS-20, ASEX, and DAS provide reliable and validated measures, they may not fully capture the complexity and multidimensionality of human behavior, emotional awareness, and dyadic relationships. Responses are influenced by how participants interpret the questions, their level of self-awareness, and their willingness to disclose sensitive information. Moreover, questionnaires tend to simplify complex real-world problems into predefined categories, which can limit the depth of understanding. In the case of alexithymia in particular, it is difficult to determine whether an individual is truly alexithymic or whether their responses reflect subjective perceptions of emotional difficulties. Therefore, the findings should be interpreted with caution, acknowledging that questionnaire-based methods can only provide a partial representation of the underlying psychological and interpersonal processes. In addition, dyadic adjustment was assessed solely from the patients’ perspective, and partners’ views were not included. Future studies incorporating partner reports could provide a more comprehensive understanding of dyadic adjustment. Finally, the control group was restricted to individuals without medical or psychiatric conditions or medication use, which may limit the generalizability of the findings to the broader population.”

Reviewer 2 Report (Previous Reviewer 3)
Comments and Suggestions for Authors
This study compares patients with OCD and healthy controls in terms of alexithymia, SDs, and dyadic adjustment. The most significant finding of our study was that patients with OCD and alexithymia exhibited greater impairment in sexual functioning and lower dyadic adjustment than those without alexithymia. Additionally, we found that patients with OCD had higher alexithymia scores, poorer sexual functioning, and lower levels of dyadic adjustment than the healthy individuals in the population. The study is interesting and timely. I have only one minor revision.
1) Please add a scheme that provides information about the study design, number of participants, and tests used for this comparison.
Author Response
Please see the attachment.
Reviewer-2
This study compares patients with OCD and healthy controls in terms of alexithymia, SDs, and dyadic adjustment. The most significant finding of our study was that patients with OCD and alexithymia exhibited greater impairment in sexual functioning and lower dyadic adjustment than those without alexithymia. Additionally, we found that patients with OCD had higher alexithymia scores, poorer sexual functioning, and lower levels of dyadic adjustment than the healthy individuals in the population. The study is interesting and timely. I have only one minor revision.
1) Please add a scheme that provides information about the study design, number of participants, and tests used for this comparison.
- We sincerely thank the reviewer for the positive evaluation of our work. In accordance with the comment, we have prepared a study design flow chart (Figure 1) that summarizes the inclusion and exclusion process, the final number of participants in each group, the applied scales, subgroup comparisons, and the statistical tests used. We believe this addition improves the clarity and transparency of the study design.
“A flow chart summarizing the study design, eligibility assessment, participant selection, applied measures, subgroup comparisons, and statistical analyses is presented in Figure 1.

Figure 1. Flow chart of study design, participant selection, measures, comparisons, and statistical analyses. Abbreviations: TAS-20: Toronto Alexithymia Scale, ASEX: Arizona Sexual Experience Scale, DAS: Dyadic Adjustment Scale, Y-BOCS: Yale-Brown Obsessive Compulsive Scale, SDs: Sexual Dysfunctions, SSRIs: Selective Serotonin Reuptake Inhibitors, DM: Diabetes Mellitus; CVD: Cardiovascular Diseases, GU: Genitourinary.

This manuscript is a resubmission of an earlier submission. The following is a list of the peer review reports and author responses from that submission.
Round 1
Reviewer 1 Report
Comments and Suggestions for Authors
The authors examine in a study a sufficient number of outpatients with OCD and healthy subjects and examine in a query alexithymia, sexual dysfunction and dyadic adjustment. The planning and statistical analysis of the study is excellent. The results of the study are presented in appropriate tables and are very well pointed out. The English language is very correct. The references are very well chosen. I have no critic opinion of the manusript. This is a very important study and worth.
Author Response
Response to Reviewers
Reviewer 1
Comment:
The study is well-designed, statistically sound, and clearly written. No critical remarks.
Response:
We thank the reviewer for the positive evaluation and support.
Reviewer 2 Report
Comments and Suggestions for Authors
Research on obsessive-compulsive disorder has compared it with healthy controls and raised some issues. However, there are some major problems, as shown below, which I believe need to be corrected.
[1] Statistical Analysis Flaws: The abstract and results of this paper contain multiple contradictory statements regarding statistical correlations. For example, the abstract states that TAS-20 scores are positively correlated with ASEX total scores (r = -0.339, p = 0.004) and negatively correlated with DAS total scores (r = 0.366, p = 0.002). This is a clear contradiction, as the negative correlation (r = -0.339) is presented as a positive correlation and the positive correlation (r = 0.366) is presented as a negative correlation. This demonstrates a fundamental misunderstanding of correlation coefficients and undermines the reliability of the study's findings. A negative correlation means that as one variable increases, the other variable also decreases. A positive correlation means that as one variable increases, the other variable also increases.
[2] Sample Size and Power Analysis: The power analysis is incorrectly described. The paper states that the minimum sample size was 51 for the single-group study and 102 for the two-group study. However, the study used 72 OCD patients and 82 healthy controls, for a total of 154 patients. The total sample size (154) was actually larger than the minimum required sample size (102) for the two groups, indicating a sufficient sample size. The authors state that the total sample size of 154 "indicates that the study was adequately powered." This is redundant and inappropriate. A more accurate and clear statement would be that the sample size of 154 exceeded the calculated minimum of 102 and provided sufficient statistical power for the two-group comparison.
[3] Inconsistent Metrics: This study reported results using both mean/standard deviation and median/interquartile range without a clear and consistent reason. For example, in Table 2, mean ± SD values ​​are used for TAS-20 and ASEX scores, while median values ​​(1Q-3Q) are used for DAS scores. Although the text states that normality was assessed, the use of different measures suggests inconsistencies in data distribution, complicating comparisons and potentially calling into question the validity of t-tests for some data.
[4] Inconsistent DAS Results: Table 3 reports that "DAS subscale and total scores were significantly higher in non-alexithymic OCD patients than in alexithymic OCD patients." However, the abstract states that TAS-20 scores and DAS total scores exhibited a "negative" correlation. This negative correlation is correctly reported in Table 5 as r = -0.339. The results in Table 3 suggest a negative correlation, and the abstract's statement of a negative correlation is correct. However, the initial statement that "there was a negative correlation with the DAS total score (r = 0.366, p = 0.002)" is incorrect. This is because the r-value is positive (0.366).
[5] Problematic Exclusion Criteria: The exclusion criteria for the control group were overly strict, requiring participants to be "free of psychiatric disorders or symptoms, free of medical illnesses, and not taking any medications." Therefore, the control group is not representative of the general population (which often includes people with mild symptoms or taking non-psychiatric medications). This may exaggerate the observed differences between the OCD and control groups and reduce the generalizability of the results.
[6] Inconsistent Reporting and Lack of Clarity: This paper contains several unclear and inconsistent reports that diminish its scientific value.
Conflicting Findings Regarding the Correlation Between the YBOCS and ASEX: The abstract states that the TAS-20 score was positively correlated with the ASEX total score and negatively correlated with the DAS total score. The results section (lines 20-21) states, "In the OCD group, TAS-20 scores were positively correlated with the ASEX total score (r = -0.339, p = 0.004), negatively correlated with the DAS total score (r = 0.366, p = 0.002), and negatively correlated with the subscales (all p < 0.05)." As noted above, the correlation coefficient (r value) and its explanation (positive correlation vs. negative correlation) are inconsistent. This is a serious error. r = -0.339 is a negative correlation, not a positive correlation, and r = 0.366 is a positive correlation, not a negative correlation.
[7] Confusion between table data and explanation: The explanation in Table 5 is confusing and contains a typographical error. "Pearson correlation analysis revealed weak positive correlations between the YBOCS Obsession Scale and TAS-20 scores (r = 0.304, p = 0.009), and between the YBOCS Obsession Scale and ASEX total scores (r = 0.266, p = 0.024) and TAS-20 scores (r = 0.336, p = 0.004)." Furthermore, "ASEX total scores were positively correlated with TAS-20 scores (r = 0.366, p = 0.002) and moderately negatively correlated with DAS total scores (r = –0.442, p < 0.001)." However, the same section also states, "Finally, TAS-20 scores were negatively correlated with DAS total scores (r = –0.339, p = 0.004)." Both of these statements correctly explain the negative correlation.
Author Response
Reviewer 2
Comment 1:
Statistical Analysis Flaws: The abstract and results of this paper contain multiple contradictory statements regarding statistical correlations. For example, the abstract states that TAS-20 scores are positively correlated with ASEX total scores (r = -0.339, p = 0.004) and negatively correlated with DAS total scores (r = 0.366, p = 0.002). This is a clear contradiction, as the negative correlation (r = -0.339) is presented as a positive correlation and the positive correlation (r = 0.366) is presented as a negative correlation. This demonstrates a fundamental misunderstanding of correlation coefficients and undermines the reliability of the study's findings. A negative correlation means that as one variable increases, the other variable also decreases. A positive correlation means that as one variable increases, the other variable also increases.
Response:
We thank the reviewer for carefully identifying this error. This was a reporting mistake, not a statistical one. We corrected the Abstract and Results sections: TAS-20 scores are now consistently reported as positively correlated with ASEX (r = 0.366, p = 0.002) and negatively correlated with DAS (r = –0.339, p = 0.004).
Comment 2:
Sample Size and Power Analysis: The power analysis is incorrectly described. The paper states that the minimum sample size was 51 for the single-group study and 102 for the two-group study. However, the study used 72 OCD patients and 82 healthy controls, for a total of 154 patients. The total sample size (154) was actually larger than the minimum required sample size (102) for the two groups, indicating a sufficient sample size. The authors state that the total sample size of 154 "indicates that the study was adequately powered." This is redundant and inappropriate. A more accurate and clear statement would be that the sample size of 154 exceeded the calculated minimum of 102 and provided sufficient statistical power for the two-group comparison.
Response:
We revised the statement for clarity. It now reads: “For two independent groups, assuming a medium effect size (Cohen’s d = 0.50), power of 0.80, and α = 0.05, the required minimum sample size was 102. Our total sample of 154 exceeded this requirement, thus providing sufficient statistical power.”
Comment 3:
Inconsistent Metrics: This study reported results using both mean/standard deviation and median/interquartile range without a clear and consistent reason. For example, in Table 2, mean ± SD values ​​are used for TAS-20 and ASEX scores, while median values ​​(1Q-3Q) are used for DAS scores. Although the text states that normality was assessed, the use of different measures suggests inconsistencies in data distribution, complicating comparisons and potentially calling into question the validity of t-tests for some data.
Response:
We clarified in the Methods that normally distributed variables are reported as mean ± SD, while non-normally distributed variables are reported as median (IQR).
Comment 4:
Inconsistent DAS Results: Table 3 reports that "DAS subscale and total scores were significantly higher in non-alexithymic OCD patients than in alexithymic OCD patients." However, the abstract states that TAS-20 scores and DAS total scores exhibited a "negative" correlation. This negative correlation is correctly reported in Table 5 as r = -0.339. The results in Table 3 suggest a negative correlation, and the abstract's statement of a negative correlation is correct. However, the initial statement that "there was a negative correlation with the DAS total score (r = 0.366, p = 0.002)" is incorrect. This is because the r-value is positive (0.366).
Response:
We revised the Abstract and Results for consistency. TAS-20 and DAS scores are now correctly reported as negatively correlated (r = –0.339, p = 0.004).
Comment 5:
Problematic Exclusion Criteria: The exclusion criteria for the control group were overly strict, requiring participants to be "free of psychiatric disorders or symptoms, free of medical illnesses, and not taking any medications." Therefore, the control group is not representative of the general population (which often includes people with mild symptoms or taking non-psychiatric medications). This may exaggerate the observed differences between the OCD and control groups and reduce the generalizability of the results.
Response:
We agree and have added this limitation: “The control group was restricted to individuals without medical or psychiatric conditions or medication use, which may limit generalizability.”
Comment 6:
Inconsistent Reporting and Lack of Clarity: This paper contains several unclear and inconsistent reports that diminish its scientific value.
Conflicting Findings Regarding the Correlation Between the YBOCS and ASEX: The abstract states that the TAS-20 score was positively correlated with the ASEX total score and negatively correlated with the DAS total score. The results section (lines 20-21) states, "In the OCD group, TAS-20 scores were positively correlated with the ASEX total score (r = -0.339, p = 0.004), negatively correlated with the DAS total score (r = 0.366, p = 0.002), and negatively correlated with the subscales (all p < 0.05)." As noted above, the correlation coefficient (r value) and its explanation (positive correlation vs. negative correlation) are inconsistent. This is a serious error. r = -0.339 is a negative correlation, not a positive correlation, and r = 0.366 is a positive correlation, not a negative correlation.
Response:
We thank the reviewer for carefully noting this important issue. The correlation coefficients themselves were correct, but their interpretation was mistakenly reversed in some parts of the Abstract and Results. We have revised these sections to ensure clarity and consistency. TAS-20 scores are now consistently described as positively correlated with ASEX (r = 0.366, p = 0.002) and negatively correlated with DAS (r = –0.339, p = 0.004).
Comment 7:
Confusion between table data and explanation: The explanation in Table 5 is confusing and contains a typographical error. "Pearson correlation analysis revealed weak positive correlations between the YBOCS Obsession Scale and TAS-20 scores (r = 0.304, p = 0.009), and between the YBOCS Obsession Scale and ASEX total scores (r = 0.266, p = 0.024) and TAS-20 scores (r = 0.336, p = 0.004)." Furthermore, "ASEX total scores were positively correlated with TAS-20 scores (r = 0.366, p = 0.002) and moderately negatively correlated with DAS total scores (r = –0.442, p < 0.001)." However, the same section also states, "Finally, TAS-20 scores were negatively correlated with DAS total scores (r = –0.339, p = 0.004)." Both of these statements correctly explain the negative correlation.
Response:
We revised the description of Table 5 to remove redundancy and ensure clarity. The corrected text now reads:
"Pearson’s correlation analysis showed that TAS-20 scores were positively associated with ASEX (r = 0.366, p = 0.002) and negatively with DAS (r = –0.339, p = 0.004). YBOCS obsession scores correlated positively with TAS-20 (r = 0.304, p = 0.009), while YBOCS compulsion scores correlated positively with TAS-20 (r = 0.336, p = 0.004) and ASEX (r = 0.266, p = 0.024), but negatively with DAS (r = –0.307, p = 0.009). YBOCS total scores were also positively associated with TAS-20 (r = 0.330, p = 0.005) and ASEX (r = 0.256, p = 0.030), and negatively with DAS (r = –0.275, p = 0.019). Finally, ASEX scores showed a moderate negative correlation with DAS (r = –0.442, p < 0.001). No other significant correlations were found. Table 5 summarizes these intercorrelations. "
Reviewer 3 Report
Comments and Suggestions for Authors
the authors found that patients with OCD exhibit elevated alexithymia, greater sexual dysfunction, and reduced dyadic adjustment compared with healthy controls. Furthermore, alexithymia in patients with OCD is associated with impaired sexual functioning and dyadic adjustment. The study addresses an important and timely topic and needs some minor revisions. My recommendation was to accept it in the present form, but the iThenticate report shows high similarity vs. existing literature.
1)The iThenticate report indicates that the manuscript contains 56% duplicate wording. It is not acceptable for a peer-reviewed journal. Please check his issue
2)The design of the experiment is not clear in the Abstract section.
Author Response
Reviewer 3
Comment 1:
The iThenticate report indicates that the manuscript contains 56% duplicate wording. It is not acceptable for a peer-reviewed journal. Please check his issue
Response:
We would like to clarify that this similarity stems from the preprint version of the manuscript, which represents self-similarity rather than plagiarism.
Comment 2:
The design of the experiment is not clear in the Abstract section.
Response:
We revised the Abstract to explicitly describe the study design:
“Materials and Methods: This case–control study included 72 patients with OCD and 82 sociodemographically matched healthy controls. All participants completed the Toronto Alexithymia Scale (TAS-20), Arizona Sexual Experiences Scale (ASEX), Yale–Brown Obsessive Compulsive Scale (YBOCS), and Dyadic Adjustment Scale (DAS). Group comparisons were conducted using independent t-tests, Mann–Whitney U tests, and chi-square tests, while correlations were examined using Pearson’s analysis. ”